# Radiotherapy-Mediated Immunomodulation and Anti-Tumor Abscopal Effect Combining Immune Checkpoint Blockade

**DOI:** 10.3390/cancers12102762

**Published:** 2020-09-25

**Authors:** Xinrui Zhao, Chunlin Shao

**Affiliations:** Institute of Radiation Medicine, Shanghai Medical College, Fudan University, No.2094 Xie-Tu Road, Shanghai 200032, China; 19111140003@fudan.edu.cn

**Keywords:** radiotherapy, anti-tumor abscopal effect, immunomodulation, immune checkpoint, immunotherapy

## Abstract

**Simple Summary:**

Combination of radiotherapy and immunotherapy to antagonize tumors is one of hotspot issues currently. Immunotherapy can effectively alleviate the non-lasting effect of radiotherapy and tumor recurrence. Local radiotherapy can mobilize immune cells into tumor microenvironment and make tumors unable to escape from immune supervision. Therefore, the radio-immunotherapy achieves better curative effects on cancer patients. We have reviewed the regulation of multiple immune cells with different functions by radiotherapy, as well as the mechanism research and clinical outcomes combined with immune checkpoint inhibitors for cancer treatment, which provides new insights in the treatment strategies for certain types of cancer patients in clinic.

**Abstract:**

Radiotherapy (RT) is a conventional method for clinical treatment of local tumors, which can induce tumor-specific immune response and cause the shrinkage of primary tumor and distal metastases via mediating tumor infiltration of CD8+ T cells. Ionizing radiation (IR) induced tumor regression outside the radiation field is termed as abscopal effect. However, due to the mobilization of immunosuppressive signals by IR, the activated CD8+T cells are not sufficient to maintain a long-term positive feedback to make the tumors regress completely. Eventually, the “hot” tumors gradually turn to “cold”. With the advent of emerging immunotherapy, the combination of immune checkpoint blockade (ICB) and local RT has produced welcome changes in stubborn metastases, especially anti-PD-1/PD-L1 and anti-CTLA-4 which have been approved in clinical cancer treatment. However, the detailed mechanism of the abscopal effect induced by combined therapy is still unclear. Therefore, how to formulate a therapeutic schedule to maximize the efficacy should be took into consideration according to specific circumstance. This paper reviewed the recent research progresses in immunomodulatory effects of local radiotherapy on the tumor microenvironment, as well as the unique advantage for abscopal effect when combined with ICB, with a view to exploring the potential application value of radioimmunotherapy in clinic.

## 1. Introduction

The impact of radiotherapy (RT) on the non-radiation field tumor caused by the immune response triggered by the local irradiated tumor and the combination of RT and immunotherapy to eliminate the tumor have become the focus of current research. As we all know, systemic RT can ablate the body′s immune system to lower the risk of xenotransplantation rejection in patients [1]. Conversely, a volume of studies have proved that local tumor irradiation can enhance the immune response, thereby allowing metastases away from the treatment site to be controlled [2]. The previous research indicated that the local RT of a tumor increased the expression level of major histocompatibility complex (MHC) class I molecules and improved the ability of presenting antigens of antigen presenting cells (APCs), the maturation of dendritic cells (DCs) was motivated, and the toxicity of nature killer (NK) cells and the tumor infiltration of CD8+ cytotoxic T lymphocytes (CTLs) were enhanced [3,4,5]. However, at the same time, ionizing radiation (IR) also mobilized some immunosuppressive cells, such as regulatory T (Treg) cells and M2 macrophage and myeloid-derived suppressor cells (MDSCs). The presence of these cells helped the tumor cells escape the supervision of the immune system, which greatly reduced the efficiency of IR treatment.

Targeting therapies against programmed cell death protein 1 (PD-1) and its ligand PD-L1, T-lymphocyte-associated antigen 4 (CTLA-4) has impelled the clinical treatment of cancer to move forward and has also been found to extend the response of RT on the antineoplastic function to the whole body through activating T cells. Building on this foundation, the combination of RT with immune checkpoints blockade (ICB) is increasingly used in the study of local RT and the RT-induced abscopal effect [6,7,8]. Monoclonal antibodies targeting CTLA-4 (ipilimumab), PD-1 (e.g., pembro-lizumab and nivolumab), and PD-L1 (e.g., avelumab, atezolizumab, and durvalumab) have been approved by the Food and Drug Administration of USA (FDA) in the treatment of various malignant tumors, such as melanoma, non-small cell lung cancer, renal cell carcinoma, urothelial carcinoma, and lymphoma [9]. Despite the great success of anti-CTLA-4 and anti-PD-1/PD-L1 therapy, due to the complex adjustment of anti-tumor immunity in the tumor microenvironment, only a small percentage of patients benefit from ICB. Therefore, a more in-depth exploration of ICB therapy is required.

Here, we aim to discuss how local irradiation of a tumor modulates the immune/inflammatory system and mobilizes the immune cells via initiating the relevant signal transduction, how to regulate immune cell populations in the tumor microenvironment, as well as the influence and functional mechanism of ICB on local radiation fields and distant metastatic tumors. This is integrated with the published studies to evaluate the preponderance of RT combined with immunotherapy in the anti-tumor effect and to achieve the goal of maximizing the therapeutic advantages of radio-immunotherapy on the clinical treatment of cancer and metastases.

## 2. Modulation of Radiotherapy on Immune Cell Population in Tumor Microenvironment

### 2.1. T Lymphocytes

Radiotherapy affecting tumors by mediating the immune system is a new perspective proposed in cancer treatment in recent years. According to previous studies on tumor-bearing mouse models, CD8+ cytotoxic T lymphocytes (CTL) carry a major function in the local radiotherapy of tumors. The findings indicated that under a dose of irradiation, with the shrinking of the tumor volume, the number of infiltrating T cells in the tumor microenvironment increased, nevertheless, the tumor developed to be radioresistant after CD8+ T cells deleted, which reversed the growth inhibition of the tumor [10,11,12]. There is evidence that after infection of mice with PR8-H1N1 influenza virus antigens intranasally, activated T cells were detected not only distributed in the lungs, but also docked in other epithelial organs at the distant sites [13]. DCs were administered intratumorally after irradiating the femur of the mice inoculated with squamous cell carcinoma, and the tumor growth of the femur and unirradiated chest was obviously inhibited [14]. When the immune system is activated, activated T cells are evenly distributed at each metastatic site to achieve a response to distant metastases.

How are T cells activated to affect the tumor microenvironment and achieve an abscopal response? In summary, local RT of the tumor increases the expression level of major histocompatibility complex (MHC) class I molecules and improves the ability of presenting antigens of antigen presenting cells (APCs), promotes the maturation of DCs, and initiates T cell activation, which in turn raises the level of CD8+ T cells (Figure 1) [3,15].

In addition, local RT also triggers the process termed “immunogenic cell death” (ICD) to induce the cell death, a process that stimulates the release of endogenous damage-associated molecular patterns (DAMPs) including calreticulin, high-mobility group box 1 protein (HMGB1), and adenosine triphosphate (ATP), which primes the immune system and promotes the presentation of antigens to T cells (Figure 1). Specially, HMGB1 plays a pro-inflammatory mediator role in producing inflammatory factors TNF, IL-1, IL-6, and IL-8 by stimulating monocytes, as well as binding to toll-like receptor 4 (TLR4) on DCs and decreasing the degradation of intracellular antigens to facilitate the presentation of tumor antigens [5,16,17]. Subsequently, the mature DCs move to the lymph nodes to perform the function of activating T lymphocytes [5]. The administration of Fms-like tyrosine kinase receptor 3 ligand (Flt3-L), which activates DCs, can retard the growth of spontaneous metastasis (67NR)-mediating T cells activity under RT treatment in the murine model of lung cancer [18]. Tumor cells release multiple chemokines after receiving radiation, including CXCL9, CXCL10, and CXCL16, and these chemokines can recruit activated T cells to tumors [1,19]. Costimulatory molecules (CD80), intercellular adhesion molecule 1 (ICAM-1), stress ligands (NKG2DL), and death receptor Fas on the tumor surface are also upregulated after radiation, which makes the tumor cells more sensitive to the recognition and killing of cytotoxic T lymphocyte (Figure 1) [16,20,21,22]. Activated T cells flow through the body with blood and lymph.

For another T cell, named the T helper (Th) cell, also known as CD4+ T cell, it has been proved to induce differentiation in the presence of IR [23]. According to the type of cytokines secreted from Th cells, the Th function is zoned as Th1, Th2, Th17, and regulatory T (Treg) cells. Briefly, Th1 cells secrete IL-2 and IFN-γ mostly, which can enhance the killing ability of NK cells, promotes the differentiation of macrophages, and aggravates the activation of cytotoxic T cells, thereby effectively mediating the cellular immune process. IL-4, IL-6, and IL-10 are mainly secreted by Th2 cells whose key function is to mediate humoral immunity and stimulate antibody production. Th17 cells produce IL-17, IL-21, IL-22, and the master transcription factor RORγτ, responsible for the eradication of intracellular pathogens and fungi. In contrast, Treg cells play an immunosuppressive role by mediating TGF-β secretion (Figure 1). The salient marker of Treg cells is CD4+CD25+ as well as the surface marker of fork-head box protein 3 (Foxp3) [24,25,26]. Compared with other T cells, Treg cells exhibit obvious radiation resistance, which can be verified in several studies whereby radiotherapy increases the proportion of Treg cells in the immune microenvironment but not in the draining lymph nodes, and the increase of the ratio of Treg/effector-T cell attenuates the effect of radiotherapy to a certain extent [27,28,29,30]. In view of this, Treg cell ablation combined with radiotherapy has been extensively investigated for improving the clinical outcome of tumor therapy [27,31,32]. It is also reported that in the early stage of pneumonia, local irradiation of the chest cavity amplified the proportion of CD4+ CD25+ FoxP3+Treg, which may prevent T cells from over-activating the immune response to cause lung damage and re-establish the homeostasis of the body [33]. On the other hand, radiotherapy can stimulate the complex of tumor antigen and MHC-II to be presented to CD4+ T helper cells through DC (Figure 1), enhancing the response of CD8+ CTL to tumor cells [34,35].

### 2.2. Nature Killer Cells

In tumor immunity, in addition to CD8+ T cells with remarkable efficacy in tumor killing, other immune cells also play an important role likewise, and NK cells are one of them. NK cells are large granular lymphocytes involved in the elimination of pathogens and tumor cells in innate immunity. Several inhibitory receptors (e.g., KIR) and activated receptors (e.g., NKG2D, CD16) were expressed on the surface of NK cells (Figure 1). The activation of NK cells depends on the balance between activation and suppression signals [36,37].

In vitro experiments testified that IR synergized with a histone deacetylase inhibitor increased the expression of NKG2D in human colorectal cancer K12 cells, cervical carcinoma HeLa cells, and melanoma A375 cells [38]. IR can also induce the upregulation of NKG2D ligands (NKG2DLs) to activate the NK pathway. However, there are reports of adverse reactions whereby IR could elevate the expression of matrix metalloproteinases (MMPs) proteins related to tumor metastasis so that NKG2DL fell off the tumor surface, which leads the tumor to escape from the surveillance of NK cells. When IR was combined with MMP inhibitors, the expression of NKG2DL obviously increased and facilitated cancer cells sensitive to the killing effect of NK cells [39]. Furthermore, NK cells can also be inactivated by the exposure of RT-driven MHC-I molecules on the surface of tumor cells, but the final response of NK cells depends on who gains the upper hand in the activation and inhibition functions.

### 2.3. Macrophage

Tumor-associated macrophages (TAMs) are regarded as important immune cell components in the tumor microenvironment, and they are myeloid cells abundant in the stromal cavity of diverse solid tumors [40,41]. Notable for the functional plasticity and heterogeneity of cells, TAMs can differentiate into different phenotypes in the presence of various stimuli, such as the regulation of cytokines, growth factors, nucleotides, microbial products, and many other factors [42]. Macrophages differentiate into a cytotoxic activation status with M1 phenotypes when stimulated by interferon-γ (IFN-γ), tumor necrosis factor-α (TNF-α), or microbial products such as lipopolysaccharide (LPS) (Figure 1) [42,43]. Among them, IFN-γ is a major cytokine and a critical product of Th1 cells that is extremely related to M1 macrophage activation, and the receptor of IFN-γ on the cell surface is composed of IFNGR-1 and IFNGR-2 chains [44]. The combination expression of IL-12^high^IL-23^high^IL-10^low^ is identified as the phenotype of M1 macrophage with increased expression of several pro-inflammatory cytokines, such as IL-1β, IL-6, and TNF-α [45]. In these macrophages, the downstream signals of IFNs and toll-like receptors (TLRs) are transduced by activating the signal transducer and activator of transcription 1 (STAT1) and nuclear factor-kB (NF-kB), driving the transcription and expression of some genes including chemokines C-C motif ligand 15 (CCL15), C-X-C motif ligand 10 (CXCL10), chemokine receptors such as C-C chemokine receptor type 7 (CCR7), and reactive oxygen species (particularly inducible nitric oxide synthase, iNOS) [43,45]. Another type of macrophage is classified as M2, which characterized by tissue repair, secretion of immunosuppressive cytokines, angiogenesis, and promotion of tumor progression according to its source of stimulation, the difference in receptors and signal transduction pathways. Through the primary stimulus, Th2 related cytokines (such as IL-4 or IL-13), IL-10, immune complexes, glucocorticoid activation and companied with an IL-12^low^IL-23^low^IL-10^high^TGF-β^high^ phenotype. The recognition of the M2 macrophage is the expression of immunosuppressive cytokines, chemokines, and surface markers (such as IL-10, CCL17 and CD206.9) with elevated expression of arginase 1 (Arg-1), growing secretion of some other factors related to wound healing such as colony stimulating factor 1 (CSF1), vascular endothelial growth factor (VEGF), and IL-8 [43,46,47].

In a group of experiments, peritoneal macrophages of unirradiated mice expressed M2-related transcription factors with a small amount of iNOS. In stark contrast to the unirradiated group, HIF-1, Ym-1, Fizz-1, and arginase expression decreased and iNOS expression increased after receiving 2 Gy of systemic irradiation, indicating the acquisition of a radiation-induced M1 phenotype [48]. However, some other results concluded that in the oral cancer model of mice, IR boosts the infiltration of macrophage in the tumor, causing the anti-inflammatory phenotype M2 to exert and cumulate in the hypoxic site of the tumor after radiation [49,50]. Overall, RT can activate the differentiation of M1 macrophages and promotes M1 macrophage flow into tumor cells and prevents the conversion to M2 type to ensure the therapeutic effect [51,52,53]. RT-reprogrammed macrophages have a profound effect on oncotherapy, and the conversion of M2 to M1 phenotype promotes tumor treatment and serves as an implicit mediator of distal effects.

### 2.4. Myeloid-Derived Suppressor Cells

Apart from the already referred to Treg and M2 macrophages, MDSCs are also a type of cell that plays an immunosuppressive role in the tumor microenvironment. MDSCs are the aggregation of heterogeneous cells of myeloid origin, and the maturation period of MDSCs in the tumor microenvironment is relative longer, hence IR may cause a different impact on the immunomodulation of MDSCs [54]. In the late 1990s, CD11b+Gr1+ was identified as the phenotype of MDSCs in a mouse model [55].

Depending on the different cancer models, fractionated irradiation performs a dual influence of MDSCs in tumor progression by recruiting, removing, repolarizing, and reorganizing MDSCs and inducing antigen presentation [56]. For the effect of tumor-promotion, IR stimulated the secretion of the granulocyte–macrophage colony-stimulating factor (GMC-SF) and prompted the movement of MDSCs to the circulatory system and inflammatory tissues [56]. In a prostate tumor-bearing mouse model, when the primary tumor site was irradiated with a fraction dose of 3 Gy×5, a certain increase in MDSCs in the spleen, lung, lymph nodes, and peripheral blood was detected. This induction may be induced by DNA damage, which triggers the translocation of the kinase ABL1 to the nucleus and binds to the CSF1 promoter to enhance its transcription, then further mediates the infiltration of MDSCs into the tumor (Figure 1) [57]. In contrast, an array assay has demonstrated the superiority of RT combined with immunotherapy to thwart MDSCs and optimize CD8+ anti-tumor. In patients with limited oligometastatic diseases (OMDs) treated with stereotactic body radiotherapy (SBRT) and sunitinib (a multitargeted receptor tyrosine kinase inhibitor), the therapeutic effect is positively correlated with a decrease in the number of mononuclear MDSCs and Treg and B cells and the increase of Tbet expression in primary CD4 and CD8 T cells [58]. Analogously, the combined use of anti-PD-L1 and irradiation (12 Gy) in the mice model inoculated subcutaneously with TUBO (mammary tumor cells) and MC38 (colon adenocarcinoma cell line) induced apoptosis of MDSCs to limit their accumulation in the tumor microenvironment and restored the anti-tumor function of CD8, which in return eliminated MDSCs [59].

## 3. Impact of Radiotherapy Combined with Immune Checkpoint Blockade on Abscopal Effect

### 3.1. PD-1/PD-L1 Blockade

Since the first PD-1 drug was approved by the Food and Drug Administration (FDA) in 2014, the application of PD-1/PD-L1 inhibitor has produced a revolutionary breakthrough in clinical cancer treatment [60,61]. Based on this, a volume of studies have demonstrated the emerging use of radiotherapy combined with ICB in investigating primary tumor progression and anti-tumor abscopal effect. What is the significance of targeting the PD-1/PD-L1 axis? Simply put, PD-1 is expressed on the activated T cells, NK cells and B lymphocytes, its ligand PD-L1 (B7-H1, CD274) is extensively expressed on T cells and endothelial cells and is highly expressed in different types of tumors, the interaction between PD-1 and PD-L1 results in the exhaustion of T cells, restrains the release of cytokines and the killing ability of T cells, and further triggers the immune escape of tumors [62,63]. Therefore, PD-1/PD-L1 blockade can enhance the immune surveillance of tumors and reverse the anti-tumor efficacy (Figure 1).

A number of cases have confirmed that RT combined with anti-PD-1/PD-L1 can effectively control the occurrence of tumors outside the radiation field. In the mouse model, renal carcinoma cells were inoculated as primary and secondary tumors and breast cancer cells were injected as third tumors. The primary tumors were treated with local RT at a single dose of 15 Gy and anti-PD-1 administration, and the growth of a secondary tumor was obviously controlled, but the lateral breast tumor did not respond to the treatment, implying that the abscopal effect was tumor specific [64]. In the experiment of modeling brain metastasis, a melanoma was inoculated into the intracranial area and contralateral flank of mice, and a 2 Gy × 4 fractionation irradiation dose was delivered to the head combined with anti-PD-1. The results revealed that the ratio of tumor infiltrated CD8+ T cells was increased and the proportion of CD4+ Treg was reduced in the flank tumor. CD8+ T cells obtained using flow cytometry were then re-stimulated with anti-CD3/CD28 in vitro, and CD8+ cells in the combination treatment group expressed higher INF-γ and Tbet [65]. Of note, similar results have been elaborated in mouse models of lung cancer, glioblastoma, breast cancer, prostate cancer, and osteosarcoma [66,67,68,69,70]. Clinical trials have also clarified abscopal effects. Of 126 cancer patients receiving PD-1 inhibitors and RT, 53% (67/126) of patients were treated simultaneously, and 36% (24/67) patients met the inclusion criteria, that is, RT started within one month of the first or last application of PD-1 inhibitors (pembrolizumab or nivolumab) and at least one metastasis lesion outside the 10% iso-dose range of the prescribed radiation dose. The ultimate examination results showed that 29% (7/24) of patients had the abscopal effect, 43% had melanoma, 43% had non-small cell lung cancer (NSCLC), and 14% had renal cell carcinoma (RCC), respectively [71].

What are the advantages of PD-1/PD-L1 blockade combined with RT over monotherapy? Pre-clinical studies showed that after fractionation radiation, IFN-γ produced by CD8+ T cells mediated the upregulation of PD-L1 on tumor cells and induced a local anti-tumor response. Radiotherapy alone was unable to maintain long-term anti-tumor immunity, however, it could relieve the limitation of RT on immunity by blocking the PD-1/PD-L1 axis [72]. However, anti-PD-1 is largely unsuccessful at overwhelming the secondary tumors without the assistance of RT [64]. The complete regression both of irradiated and out-of-field tumors is rare in the case of fractionated RT singly. In the irradiated tumor area, the elevated expression of PD-L1 on the surface of tumor cells and CD11b+ Gr1+ cells induced by RT seemed to be the hallmark of local CD8+ T cells activation, which indicated that the immune response was confined to the irradiated tumor site. When targeting the PD-1/PD-L1 axis on the basis of local RT, it broke the limitation, activated the systemic immunity to exert a distal anti-tumor ability, and extended the expression of PD-L1 on the tumor cells and MDSCs in non-radiation lesions [73].

### 3.2. CTLA-4 Blockade

Referring to cytotoxic T lymphocyte antigen 4 (CTLA-4), bearing a likeness to PD-1/PD-L1, it has also been a high-profile target of immune checkpoint therapy in recent years. Ipilimumab, the CTLA-4 inhibitor, has been approved by the FDA for the treatment of patients with advanced melanoma [74]. A growing number of experiments have illustrated that the synergy of RT and CTLA-4 inhibitors effectively improves the survival of cancer patients [16]. CTLA-4 and CD28 are homologous receptors co-expressed on the surface of CD4+ and CD8+ T cells which can bind to the same ligands, CD80 and CD86, on the surface of APCs cells but present opposite feedback on T cell activation (Figure 1). Among them, CTLA-4 interacting with ligands is responsible for the inactivation of T cells [75,76]. Studies have confirmed that CTLA-4 has a higher affinity to ligands than CD28, which may cause the antagonism of CTLA-4 on CD28-activated T cells [75,77]. Under this premise, targeting CTLA-4 in combination with RT is of great significance for the clearance of primary tumors and distant metastases.

Employing the spontaneous metastasis mouse model bearing 4T1 mammary carcinoma cells with poor immunogenicity, in contrast to other monotherapy groups, the union of anti-CTLA-4 and local RT has statistical significance in the reduction of in situ tumor volume and lung metastases. The suppression of lung metastases positively related to the increase of survival through CD8+ T cells [78]. In a similar mammary carcinoma mouse model, IR (12 Gy × 2 fractionation dose) together with CTLA-4 blockade induced the secretion of CXCL16, which could bind to CXCR6 on the surface of Th1 cells and co-stimulate the CD8+ T cells to be recruited to the tumor sites, resulting in the regression of the primary tumor and metastases [79]. A patient suffering from NSCLC with multiple metastases was given 5 times of fractionated RT at the site of a liver metastasis with a total dose of 30 Gy and administered with ipilimumab. After the treatment, the CT scanning manifested that the liver, lung, bone, lymph nodes, and other non-radiated metastatic lesions had subsided. Meanwhile, two biomarkers of absolute lymphocyte counts (ALCs) and absolute eosinophil counts (AECs) turned out to be closely linked to the survival of melanoma patients treated with ipilimumab after combined therapy [80]. Intriguingly, in the 4T1 spontaneous metastasis mouse model, some researchers found that receiving anti-CTLA-4 alone caused rapid motility of tumor infiltrating lymphocytes (TILs) in the tumor; local IR could also slightly enhance this motility. However, when these two methods were superimposed, TILs were arrested. In this circumstance, it was believed that IR contributed to the upregulation of retinoic acid early inducible–1 (RAE-1) on the tumor surface, a ligand of NKG2D. Binding of RAE-1 and NKG2D impaired the movement of TILs. Once NKG2D was inhibited, the arrest of TILs was offset. In addition, the significant reduction of primary tumors using combined therapy may improve the inhibitory effect of TGF-β on REA-1 expression [81,82].

As the research moves along, the combination of RT with anti-CTLA-4 and anti-PD-1/PD-L1 is being used to achieve the optimal therapeutic effect. Researchers have constructed three metastatic tumor mouse models of melanoma, breast cancer, and pancreatic cancer. It was indicated that anti-CTLA-4 primed the suppression of Treg cells to broaden the ratio of CD8/ Treg, and anti-PD-1/PD- L1 mainly increased the proportion of CD8+ TILs, but only achieved a higher response with the participation of RT, which diversified the T cell receptor (TCR) of TILs from unirradiated tumors [83]. The research implies that triple therapy (RT + anti-CTLA-4 + anti-PD-1/PD-L1) in various cancer types is not redundant for cancer treatment. On the contrary, each therapy is responsible for different responsibilities of immune mobilization and mutual promotion, unfolding the superiority of triple therapy. The above cases of the effects of radiotherapy combined with immune checkpoint inhibition on abscopal tumors are summarized in Table 1.

## 4. Potential Mechanism of Radiation-Induced Abscopal Effects

### 4.1. Radiation Increases the Antigen Presentation

Radiation fortifies the exposure of tumor antigens to make them more visible to the immune system. Various mechanisms are involved in this process. The direct effect of radiation-induced DNA damage releases neo-antigens on tumors. The accumulation of mutant nucleic acid and proteins increases the odds of ICB treatment triggering tumor immunogenicity [84,85]. Besides modulating the release of antigens, the MHC molecules on tumor cells are also upregulated upon radiation, which activates the killing of T cells against tumors by binding to T cell receptors (TCRs) in the form of MHC-peptide transmitted by antigens presentation [86,87]. Furthermore, the irradiated tumor cells can also release DAMP and cytokines, which can enhance the translocation capacity of immune cells. When combined with ICB, the above effect seems to be more intense [88].

### 4.2. Radiation-Induced Activation of the cGAS-STING Signaling

In addition to the mode of antigen presentation, the production of IFN is crucial for RT-induced immune activation and abscopal response, while the immanent IFN secretion of irradiated tumor cells relies on the opening of cytosolic DNA sensor cyclic GMP-AMP (cGAMP) synthase (cGAS)-stimulator of the interferon genes (STING) pathway and tumor microenvironment [5,89]. Radiation damage triggers the release of nuclear DNA into the cytosol, and cGAS can identify the mutant DNA in the cytoplasm and catalyzes the production of cGAMP, which serves as a second messenger to activate the ER membrane adaptor protein STING, which further stimulates TBK1 to activate the downstream transcription factors IRF3 and NF-κB. Subsequently, the expression of downstream type I interferon (IFN-I) is upregulated by the activated cGAS-STING axis through IRF3/NF-κB-dependent transcriptional activation (Figure 2) [87,90,91].

Moreover, BATF3-DCs perceive the IFN secreted from irradiated tumor cells and then recruit on the tumor cells, which is indispensable for the cross-priming of T cells to exert an anti-tumor function [92,93]. DCs then transfer to the lymph nodes, motivating CD8+ T cells and inducing cytotoxic reactions. The activated CTL, NK cells, and Th cells move to the distant tumor site via the blood circulation, where they can control the growth of the unirradiated tumor by virtue of a pro-inflammatory response [94].

### 4.3. Modulation of Immune Checkpoints by Radiation-Induced Signaling

The cGAS/STING signaling occupies an important status in the radiation and ICB response. Anti-PD-L1 exhibits a tumor inhibition effect in wild-type mice but not in mice deficient in cGAS or STING. Intramuscular injection of cGMP markedly improves the antineoplastic effect of PD-L1 blockade, implying that the cGAS pathway is required for the T cell initiation by anti-PD-L1 [95]. It has been mentioned that DNA exonuclease Trex1 can be induced by radiation above 12–18 Gy in diverse types of cancers. Trex1 could degrade DNA aggregated in the cytoplasm during radiation to impair the immunogenicity of cancer cells, as well as weaken the activation of DNA sensor cGAS/STING. In a mouse model inoculated with tumor cells on the bilateral flank, the experimental data showed the local irradiation group knocked down the expression of Trex1 and the administration of anti-CTLA-4 restored the ability of abscopal effect induction, in contrast to the Trex1 upregulated group [96]. The above conclusions indicate that the abscopal effect of radiation induced by ICB depends on cGAS/STING. In addition, the IFN-dependent manner is a classic pathway to induce the expression of PD-L1. Hence, the cGAS/STING pathway is also a pivotal upstream signal. Beyond that, the expression of PD-L1 can also be regulated by the pathway of Janus kinase- signal transducer and activator of transcription- interferon regulatory factor (JAK-STAT-IRF). After binding to the IFN receptor, IFN phosphorylates JAK to initiate the JAK-STAT-IRF pathway and then binds to the promotor of PD-L1 to upregulate its expression. In consequence, IFN adjusts the PD-L1 expression via the JAK-STAT-IRF axis [97]. Recent reports clarify that PD-L1 can be upregulated in response to a DNA double strand break (DSB) dependent on the activation of ATM/ATR/Chk1 kinase [98,99]. Additionally, it was found that the activation of the epidermal growth factor receptor (EGFR) after radiation can elevate the expression of PD-L1 through the IL-6/JAK/STAT3 pathway in NSCLC [100]. Accordingly, it can be concluded that RT combined with ICB can relieve the immunosuppression, enhance the immunogenicity of tumor cells to promote the attack of T cells on the tumor, and induce systemic immunity to enhance the abscopal effect induced by RT (Figure 2).

## 5. Clinical Outcome on Combined Therapy

In many clinical cases, RT combined with ICB treatment has demonstrated considerable superiority in cancer patients (summarized in Table 2). A 65-year-old woman was diagnosed with a metastatic mucosal melanoma which was found in early 2015, and her right buccal mucosa was discolored and diagnosed as a malignant melanoma, but the neck, chest, and abdomen were negative on CT. She underwent surgical resection in August 2015 and received adjuvant intensity modulated radiotherapy (IMRT) of 50 Gy in 20 fractions at the primary site between October and November 2015. Then, metastatic melanomas in her neck and lungs were found in June and July 2016 respectively. Starting from August 2016, she received four cycles of 200 mg pembrolizumab (anti-PD-1) (3 mg/kg) intravenously every three weeks, and then her neck was irradiated with 24 Gy in 3 fractions on days 0, 7, and 21, starting from 19 October 2016. The tumors in her neck had shrunk by 20% one week after the first radiotherapy, and then it was surprisingly found that the volume and number of lung lesions had subsided. During the period, the treatment performed well until July 2017, where two lung lesions were found to be growing, and then, with continued SBRT and 20 cycles of pembrolizumab administration, the overall disease remained stable [101].

In April 2004, a 33-year-old woman was diagnosed with cutaneous melanoma, and then the primary lesion was surgically removed. During this period, she was in good health. In 2008, a new lung nodule was discovered, and then standard chemotherapy was given. In February 2009, new lung nodules were surgically removed, and the pathological examination demonstrated that they were metastatic melanomas. In August 2009, a CT scan revealed a recurrent disease accompanied by a new pleural-based spinal mass and right hilar lymphadenopathy, which was followed by a total of 4 cycles of ipilimumab (anti-CTLA-4) (10 mg/kg) injection every three weeks. After that, ipilimumab treatment was continued, once every 12 weeks. By November 2010, her pleural-based paraspinal mass and new splenic lesions had increased, so a total of 28.5 Gy doses in 3 fractions of 6-MV photons were administrated to the paraspinal mass over 7 days. In February 2011, she received ipilimumab again. After 2 months, the lesions in the target irradiation field obviously regressed. What was even more surprising was that the lesions outside the irradiation sites (right hilar lymphadenopathy and splenic lesions) also regressed [102].

In another case, a 46-year-old man was diagnosed with advanced non-keratotic nasopharyngeal carcinoma in November 2010 and then underwent a combination of radiotherapy and chemotherapy (cisplatin and fluorouracil). However, in October 2011, he first developed distant and local recurrent disease and was subsequently diagnosed with bone metastasis. Immunotherapy with pembrolizumab began in January 2016, but after 6 months, the tumor continued to spread. Thus, the recurrent lesions were re-irradiated at a total dose of 45 Gy in 25 fractions over 6 weeks, pembrolizumab was suspended during radiotherapy. 7 weeks after completion of radiotherapy; pembrolizumab treatment was started again. In December 2016, re-staging of PET-CT and MRI was performed and revealed that all the tumor lesions had regressed significantly, companied by marked swelling of the lacrimal and salivary glands, which indicated an enhanced immune response. Subsequently, pembrolizumab was used for treatment once more, and the tumor further regressed, inflammation symptoms resolved, and local control was achieved without metastasis [103].

## 6. Conclusions and Perspectives

RT plus immunotherapy has become a trend to fight against tumors, which breaks through the confined cognition in traditional radiobiology that radiation causing the cytotoxicity on tumor cells mainly originates from the production of DNA DSBs. The mobilization and tumor infiltration of CD8+ T cells are important guarantees for radiation to keep functioning [12]. After irradiation, the damaged tumor cells release neo-antigens, DAMPs, and some chemokines, which further induce CD8+ T cells priming. Among those, the antigens presentation of APCs is particularly important to activate CD8+ T cells. The activated CD8+ T cells can migrate and infiltrate the metastases outside of the irradiated field when a certain amount is reached, resulting in the subsequent sustained effects of anti-tumor [104]. Nonetheless, CD8+ T cells are not to fight a lone battle, multiple immune cells are also involved. Noticeably, some of these immunosuppressive cells can baffle the anti-tumor immunity and abscopal effects stimulated by local RT to promote the survival of tumors, including M2 macrophages, MDSCs, immature DCs, and Treg [105,106,107]. There is a balance between immunosuppression and immunostimulation. Without intervention, the function of CD8+ T cells is insufficient to completely eradicate residual tumor cells in this equilibrium, causing tumor recurrence and confining the therapeutic efficacy of local RT [11,108].

However, the accession of immune checkpoint inhibitors (ICIs) upsets the balance via enhancing CD8+ T cells’ response. Immune checkpoints act as the “brake” of immune cells, which can invalidate the effector T cell response when specifically binding to its ligand, so that the cancerous cells can be free from immune surveillance and survive [7]. Therefore, local RT plus ICB induces a cascade reaction that magnifies and spreads the positive immune response continuously, and hence, it has become a promising treatment modality for both primary tumors and distant metastases in patients [109]. It has been clinically acknowledged that the anti-PD-1/PD-L1 axis and anti-CTLA-4 are remarkable for malignant tumors. In addition, although many new ICIs including anti-LAG-3, anti-TIGIT, and anti-TIM-3 are still in clinical trials, they also show unique advantages and huge potential and have been reported to reverse the exhaustion of T cells or NK cells to restore the lethality to tumors [110,111,112].

So, how does radiation affect the abscopal effects by modulating the immune cells? As we mentioned above, radiation increases antigen presentation by mediating the release of new antigens, increases surface antigens, and promotes the release of signaling molecules and inflammatory factors. Radiation-induced DNA damage also activates the cGAS-STING signaling pathway to release IFN. Both of these patterns are very critical for the activation and migration of T cells to distant sites to induce the abscopal effect. However, the radiation can also boost the expression of immune checkpoint ligands such as PD-L1, which makes the tumor escape the attack of the immune system. Thus, ICB application is of great significance for RT-induced abscopal effects and improves the cure of cancer. Although there have been extensive studies, the specific mechanism of radiation induced abscopal effect (RIAE) still needs to be further demonstrated.

Although the effect of ICIs is remarkable, its effect on certain tumors (such as gastrointestinal cancers, breast cancer, sarcomas, and part of genitourinary cancers) is still limited. How to improve the efficacy of immune checkpoint inhibitors is a critical issue. Two strategies are currently being taken into consideration. The first strategy is to improve the response rate of ICIs in patients via specific predictive factors (PD-L1 expression, tumor mutation burden (TMB), and clinical characteristics). The other strategy is the combination of therapeutics with ICIs. Among them, the abscopal effect induced by RT combined with ICIs is frequently used to evaluate the efficacy of cancer immunotherapy. However, recent studies have also focused on the potential functions of some molecules, which can modify the immune microenvironment to up-regulate the proportion of immune-activated cells and down-regulate the immune-suppressive cells to improve the efficacy of cancer immunotherapy. The drugs include microbiota modifiers, drugs targeting co-inhibitory receptors, anti-angiogenic therapeutics, small molecules, and oncolytic viruses [113].

As the superiority of RT combined immunotherapy is gradually emerging, some of the challenges faced cannot be neglected. The optimal dose and fractionation of RT, optimum administration point provided for the two therapeutic methods, and relevant biomarkers for tumor prediction are all variable factors for the efficacy of tumor treatment [36]. At the same time, the exploration of novel ICB and immune mechanisms and the collocation of different ICIs for RT to promote abscopal effects on more types of cancers are also issues that need to be considered in the future.

## Figures and Tables

**Figure 1 cancers-12-02762-f001:**
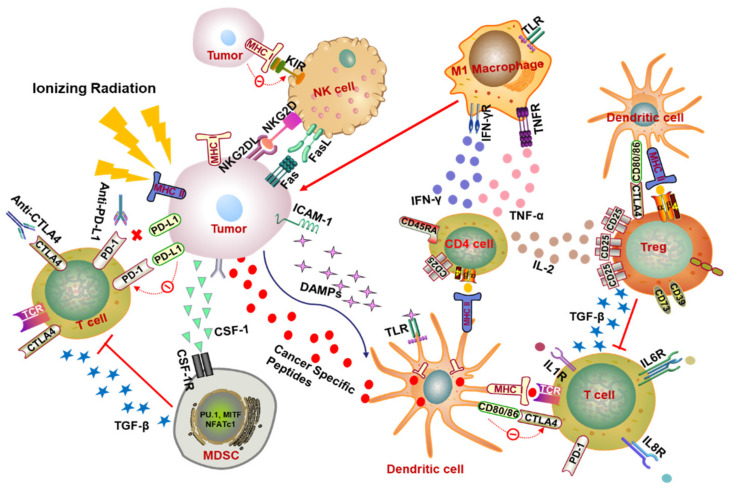
Radiotherapy-mediated immunomodulation of tumor microenvironment. In the immune activation effect, RT triggers the tumor to release DAMPs, which can bind to TLR4 on the surface of DCs to enhance the ability of DCs to present the major histocompatibility complex class I (MHC-I) molecules and promote the maturation of DCs and the activation of T cells. Irradiation also activates CD4+ T cells by increasing the presentation of MHC-II by APCs (e.g., DCs) and expedites the differentiation of M1 macrophages by secreting pro-inflammatory factors such as IFN-γ and TNF-α to promote tumor phagocytosis. On the other hand, the immunosuppressive effect also exists after radiation. Both IL-2 secreted by CD4+ T cells and MHC-II binding to Treg surface receptors contribute to the activation of immunosuppressive Treg cells. The combination of CTLA-4 and CD80/86 also promotes the activation of Treg when they have opposite effects on the activation of T cells. Subsequently, Treg exhibits an immunosuppressive role by releasing signaling factors such as TGF-β to inhibit T cell activation. Ionizing radiation (IR) can also cause the accumulation of immunosuppressive MDSCs in the tumor microenvironment by receiving CSF-1 and other factors released by the tumor and exerts its immunosuppressive function by restraining the activation of T cells. After irradiation, the expression of immune checkpoint molecules such as PD-L1 on the tumor surface will increase. Activity of T cells can be inhibited when PD-L1 binds to T cell surface receptor PD-1. However, this inhibitory effect can be relieved by the corresponding antibody anti-PD-1/PD-L1. NK cells are extremely important tumor-killing immune cells, which can directly target the tumor cells without being restricted by MHC molecules. When undergoing RT, the existence of activated receptors (e.g., KIR) and inhibitory receptors (e.g., NKG2D, CD16) causes the activation of NK cells, which depends on the balance between the activated signals and the inhibitory signals. ***Abbreviations:*** DAMPs, damage-associated molecular patterns; TLR4, toll-like receptor 4; DCs, dendritic cells; MHC, major histocompatibility complex; APCs, antigen presenting cells; PD-1, programmed cell death protein 1; PD-L1, programmed death ligand 1; CTLA-4, cytotoxic T lymphocyte antigen 4; IFN-γ, interferon-γ; Treg, regulatory T; TGF-β, transforming growth factor-β; MDSCs, myeloid-derived suppressor cells; CSF-1, colony stimulating factor-1; NK cells, nature killer cells.

**Figure 2 cancers-12-02762-f002:**
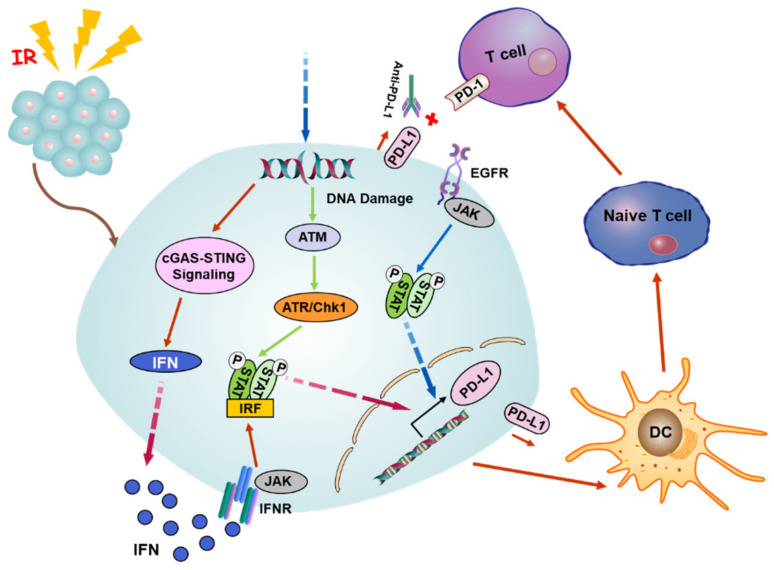
A potential mechanism of radiation-induced abscopal effect. Radiation triggers the release of nuclear DNA into the cytoplasm, and cGAS can recognize the damaged DNA and activate the cGAS-STING pathway to induce the secretion of IFN-I that can bind to the cell surface receptor INFR to activate the axis of Janus kinase/signal transducer and activator of transcription/interferon regulatory factor (JAK/STAT/IRF). Meanwhile, the DNA damage also stimulates the ATM/ATR/Chk1 signaling followed by the STAT-IRF pathway. The phosphorylated STAT/IRF complex can be translocated to the nucleus and targets the promoter of PD-L1 to boost the expression of PD-L1. Alternatively, the expression of PD-L1 is upregulated by EGFR receptor activation through JAK/STAT signaling upon radiation. The combination of anti-PD-L1 and PD-L1 breaks the connection between PD-L1 and PD-1, making tumor cells unable to escape from the immune supervision of T cells. Abbreviations: cGAS, cyclic GMP-AMP synthase; STING, stimulator of interferon genes; IFN-I, type I interferon; PD-L1, programmed death ligand 1; EGFR, epidermal growth factor receptor.

**Table 1 cancers-12-02762-t001:** Systemic effects observed in preclinical and clinical studies after RT combined with immune checkpoint inhibitor.

Inhibited Checkpoint	Tumor Type	RT Location	Treatment	Systemic Effects + Key Mediator
Preclinical Mouse Models				
PD-1	Renal carcinoma, metastatic mammary carcinoma (4T1)	right hindlimb	SABR (15 Gy) of primary tumor + anti-PD-1 mAb, i.p.	Size reduction of primary renal tumor and distal renal tumor, but not distal breast tumor, ↑CD8 + CTLs [64]
	Melanoma (D4M)	head	RT (2 Gy × 4) of primary tumor + anti-PD-1 mAb, i.p.	Growth-inhibition of irradiated and non-irradiated tumor, ↑CD8+ CTLs, ↓CD4 + Treg [65]
CTLA-4	Metastatic mammary carcinoma (4T1)	right flank	RT (12 Gy × 2) of primary tumor + anti-CTLA-4 mAb, i.p (3×)	Inhibition of lung metastases, increased survival, ↑CD8+ CTLs [78]
		right flank	RT (12 Gy × 2) of primary tumor + anti-CTLA-4 mAb, i.p. (3×)	Regression of primary tumor and metastases, ↑CD8 + CTLs [79]
PD-L1 and CTLA-4	Melanoma (B16)	right flank	RT (20 Gy) of primary tumor + anti-CTLA-4 mAb + anti-PD-L1 mAb, i.p. (3×)	Regression of primary tumor and metastases and diversification of the T cell receptor (TCR) of TILs from unirradiated tumors, ↑CD8+ CTLs, ↓CD4+ Treg [83]
	Breast cancer (TSA breast cancer cell)	right flank	RT (8 Gy × 3) of primary tumor + anti-CTLA-4 mAb + anti-PD-L1 mAb, i.p. (3×)	
	Pancreatic cancer (PDA.4662)	right flank	RT (20 Gy) of primary tumor + anti-CTLA-4 mAb + anti-PD-L1 mAb, i.p. (3×)	
**Clinical Studies**				
PD-1	Melanoma, NSCLC, RCC, head and neck cancer (*n* = 24)	primary tumor	RT + pembrolizumab or pembrolizumab	29% (*n* = 7) patients showing the abscopal effect, 43% in melanoma, 43% in NSCLC, 14% in renal cell carcinoma [71]
CTLA-4	NSCLC (*n* = 1)	hepatic metastases	RT (30 Gy in 5 fractions) + 3 cycles of ipilimumab (once in 3 weeks)	Liver, lung, bone, lymph nodes, and other non-radiated metastatic lesions subsided, improved survival [80]

Abbreviations: i.p., intraperitoneal injection; SABR, stereotactic ablative radiotherapy; CTLs, cytotoxic T lymphocytes; NSCLC, non-small cell lung cancer; RCC, renal cell carcinoma; ↑ increase; ↓ decrease.

**Table 2 cancers-12-02762-t002:** Clinical cases and systemic effect for combined RT with checkpoint inhibitor.

Case	Sex/Age	Primary Tumor	First Treatment	Systemic Effects	Second Treatment	Systemic Effects
1	Female, 65	Metastatic melanoma	IMRT (50 Gy in 20 fractions) in right buccal mucosa	Metastatic melanoma found in neck and lungs after 7 moths	IMRT (24 Gy in 3 fractions) in neck + 4 cycles of pembrolizumab (once in 3 weeks) (3 mg/kg)	The neck tumor shrank and the number of lung metastases decreased [101]
2	Female, 33	Metastatic melanoma	4 cycles of ipilimumab (once in 3 weeks) (10 mg/kg)	Pleural-based paraspinal mass and new splenic lesions gradually increased	RT (28.5 Gy in 3 fractions) in paraspinal + regular ipilimumab administration for 2 months	The lesions in the target irradiation field and outside of the irradiation site (right hilar lymphadenopathy and splenic lesions) regressed [102]
3	Male, 46	Advanced non-keratotic nasopharyngeal carcinoma	IMRT (70 Gy in 2 fractions) in cavernous sinus and 54 Gy in metastatic lymph nodes of the neck + chemotherapy (cisplatin and fluorouracil)	Bone metastasis occurred	RT (45 Gy in 25 fractions) to bony metastasis + continuing pembrolizumab	All tumorous lesions regressed significantly, inflammation symptoms resolved, and there was local control without metastasis [103]

Abbreviations: IMRT, intensity modulated radiotherapy.

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
