# Peer review of "Radiotherapy-Mediated Immunomodulation and Anti-Tumor Abscopal Effect Combining Immune Checkpoint Blockade"

_cancers, 2020, doi:10.3390/cancers12102762_

Round 1

Reviewer 1 Report

Zhao and Shao have reviewed “Radiotherapy mediated immunomodulation and anti-tumor abscopal effect combining immune checkpoint blockade”, which is a very important and informative work. The reviewer has following comments and minor concerns:

  1. The authors have done a nice job to outline the advances in this fields, with detail message and also summarized illustration of the new findings. However, the reviewer suggest authors add one more table to summarize the clinical outcomes from different studies with a little detail information for each clinical studies.
  2. The Figure 1 or Fig. 1 should be consistently used.
  3. Radiation dose should be consistently use Gy or mGy, not “cGy”.
  4. There are too many abbreviations, for which the authors many provide a table to list these abbreviations otherwise, the readers easily forget them.

Reviewer 2 Report

The review manuscript by Xinrui Zhao et al. is well written and takes recent literature and research findings into account.

I have no major issues but would like the authors to consider to include additional tables to summarize findings described in "the impact of radiotherapy combined with immune checkpoint blockade on abscopal effect" (paragraph 3) and "the clinical outcome on combined therapy" (paragraph 5) to further increase the overall quality of the review.

Reviewer 3 Report

The manuscript entitled “Radiotherapy mediated immunomodulation and anti3 tumor abscopal effect combining immune checkpoint 4 blockade” is an intrigue review on the role of combination of radiotherapy and immunotherapy to antagonize several cancers.

The Authors revised the manuscript a times. It is well written and balance, so far it should be published after some minor revisions: 

  • Abbreviations should be checked (FE: in the abstract … -PD-1/PD-L1 and anti-CLTA-4…)
  • In the text there few mistyping. Please revise and correct them.
  • The first paragraph of the introduction could be deleted. These are sentences present in several reviews and that do not center the topic.
  • Please insert abbreviation for each figure and table
  • In Figure 1, the mechanism of unmasking new epitopes with radiotherapy should be better explained.
  • In the conclusions, Authors should introduce the possibility of improve cancer immunotherapies efficacy other than abscopal effect (Authors should refer to: Longo V, Brunetti O, Azzariti A, et al. Strategies to Improve Cancer Immune Checkpoint Inhibitors Efficacy, Other Than Abscopal Effect: A Systematic Review. Cancers (Basel). 2019;11(4):539.)
